# Sandalwood Oils of Different Origins Are Active In Vitro against *Madurella mycetomatis*, the Major Fungal Pathogen Responsible for Eumycetoma

**DOI:** 10.3390/molecules29081846

**Published:** 2024-04-18

**Authors:** Shereen O. Abd Algaffar, Stephan Seegers, Prabodh Satyal, William N. Setzer, Thomas J. Schmidt, Sami A. Khalid

**Affiliations:** 1Faculty of Pharmacy, University of Science and Technology, Omdurman 14411, Sudan; phd_sh086@hotmail.com; 2University of Münster, Institute of Pharmaceutical Biology and Phytochemistry (IPBP), PharmaCampus-Corrensstrasse 48, D-48149 Münster, Germany; s_seeg03@uni-muenster.de; 3Essential Oil Science, dōTERRA International, 1248 W 700 S, Pleasant Grove, UT 84062, USA; psatyal@doterra.com; 4Department of Chemistry, University of Alabama in Huntsville, Huntsville, AL 35899, USA; wsetzer@chemistry.uah.edu

**Keywords:** mycetoma, *Madurella mycetomatis*, essential oil, sandalwood oil, *Santalum*, (*Z*)-α-santalol, (*Z*)-β-santalol, antifungal activity, drug discovery, neglected tropical disease

## Abstract

In the search for new bioactive agents against the infectious pathogen responsible for the neglected tropical disease (NTD) mycetoma, we tested a collection of 27 essential oils (EOs) in vitro against *Madurella mycetomatis*, the primary pathogen responsible for the fungal form of mycetoma, termed eumycetoma. Among this series, the EO of *Santalum album* (Santalaceae), i.e., East Indian sandalwood oil, stood out prominently with the most potent inhibition in vitro. We, therefore, directed our research toward 15 EOs of *Santalum* species of different geographical origins, along with two samples of EOs from other plant species often commercialized as “sandalwood oils”. Most of these EOs displayed similar strong activity against *M. mycetomatis* in vitro. All tested oils were thoroughly analyzed by GC-QTOF MS and most of their constituents were identified. Separation of the sandalwood oil into the fractions of sesquiterpene hydrocarbons and alcohols showed that its activity is associated with the sesquiterpene alcohols. The major constituents, the sesquiterpene alcohols (*Z*)-α- and (*Z*)-β-santalol were isolated from the *S. album* oil by column chromatography on AgNO_3_-coated silica. They were tested as isolated compounds against the fungus, and (*Z*)-α-santalol was about two times more active than the β-isomer.

## 1. Introduction

Mycetoma is a serious chronic infectious disease caused by fungi and/or bacteria. It occurs mainly in tropical and subtropical regions [1,2]. The disease is usually classified by the responsible etiologic agent into actinomycetoma, caused by bacteria (e.g., *Actinomadura* sp., *Streptomyces somaliensis*, and *Nocardia brasiliensis*), and eumycetoma, caused by fungi [3]. Several fungi are considered causative agents for eumycetoma, including about 15 genera with more than 25 species. Among them, *Madurella mycetomatis*, *Madurella grisea*, *Scedosporium boydii*, and *Leptosphaeria senegalensis* are responsible for not less than 90% of the globally reported cases of mycotic mycetoma infection [3].

Mycetoma usually manifests as an initially painless subcutaneous mass with sinus formation and a purulent discharge that contains grains, characteristic of the disease. This mass may spread from the skin to involve deep tissues and even bones, resulting in deformity, a loss of function, and destruction. The sites most frequently affected are the foot and hand, but other body parts, such as the knee, arm, leg, head and neck, thigh, and perineum, may be affected. While no age exception exists for mycetoma, it occurs more frequently in young male adults around 20–40 years old. The infection usually occurs via skin lesions, e.g., thorn picks acquired while walking barefoot on contaminated soils or during manual labor, bringing the pathogens into the dermal and subdermal tissues [1,2].

The worldwide distribution of mycetoma ranges across many tropical and subtropical regions. It is most prevalent in the “Mycetoma Belt”, roughly stretching across latitudes of 15° S to 30° N. This region engulfs many countries in which high endemicity has been reported, including Sudan, Somalia, Senegal, India, Yemen, Mexico, Venezuela, Colombia, Brazil, and others [1,2,3].

Mycetoma is mainly a poverty-associated disease occurring in poorly developed, often rural, regions of the affected countries. For this reason, it has been largely overlooked for many decades, and little or no effort has been dedicated to developing new therapies. In 2016, mycetoma was finally included by the World Health Organization on the list of neglected tropical diseases [1] and henceforth received more attention in research.

Therapy is time-consuming, prone to unwanted side effects, and often too complicated, unavailable, or expensive in endemic areas. Most unfortunately, the treatment often fails. In many cases, infections recur, and amputations are often necessary [1,2].

Itraconazole has been recommended as the first-line drug option for the treatment of eumycetoma. However, its use generally requires a long treatment duration followed by surgical excision, which could result in bone involvement and possibly relapse. Current treatment options for *M. mycetomatis* infections that include several azoles, such as itraconazole, fluconazole, posaconazole, voriconazole, and ravuconazole, as well as other classes of antifungals, including echinocandins (e.g., caspofungin, anidulafungin, micafungin), allylamines (e.g., terbinafine), and polyenes (e.g., amphotericin B), show no promising clinical outcomes. A long-awaited double-blind, randomized clinical study assessing the superiority of fosravuconazole over itraconazole finally revealed that both drugs have similar efficacy rates, with no statistically significant difference [4]. Olorofim belongs to a new class of antifungal drugs known as orotomides, with a novel mechanism of action based on inhibiting the enzyme dihydroorotate dehydrogenase, subsequently leading to the obstruction of the pyrimidine biosynthesis pathway. Olorofim exhibited much better inhibition than itraconazole when assessed against *M. mycetomatis* [5].

However, new active compounds against eumycetoma and its causative agent, *M. mycetomatis*, would be highly valuable as starting points for developing the urgently needed new, affordable, and efficacious therapeutics [1].

Natural products, and essential oils (EOs) in particular, have in some instances presented significant activity against *M. mycetomatis* [6,7]. Interestingly, sandalwood oils have exhibited antifungal activity against various pathogenic fungi [8,9]. We, therefore, set out to test an array of commercially available essential oils for their in vitro activity against *M. mycetomatis*. In this publication, we report on the results of these tests, which led to the consideration of sandalwood oils as antimycetomal agents.

## 2. Results and Discussion

### 2.1. In Vitro Activity of Diverse Essential Oils against Madurella mycetomatis

A selection of 27 commercially available essential oils (EOs) were tested in vitro for their growth-inhibitory activity against various clinical isolates of *M. mycetomatis*. The results are shown in Table 1.

Of these tested EOs, the sandalwood oil obtained from East Indian sandalwood (*Santalum album*) was by far the most active, with an average MIC value as low as 0.0039% (*v*/*v*). Conspicuously, *S. album* oil and some of its constituents had previously been shown to exhibit antifungal activity against the dermatophytic fungus *Trichophyton rubrum*, accompanied by fish toxicity and an antimitotic effect induced by the main constituent α-santalol [8]. Various sandalwood oils have also shown in vitro activity against the pathogenic fungi *Aspergillus niger*, *Candida albicans*, and *Cryptococcus neoformans* [9].

### 2.2. In Vitro Activity of Different Sandalwood Oils against Madurella mycetomatis

Due to the prominent activity of the *S. album* EO, we decided to extend the study to the EOs of other species of sandalwood (SW), namely, *S. spicatum* (Australian SW), *S. austrocaledonicum* (Pacific SW), *S. lanceolatum* (North Australian SW) and *S. paniculatum* (Hawaiian SW), all available on the market. We also included two EOs not stemming from *Santalum* species but marketed under related denominations, namely, the oils of *Amyris balsamifera* (Rutaceae, “West Indian sandalwood Oil”) and of *Barchylaena huillensis* (Asteracae; “African sandalwood Oil” or “Muhuhu Oil”).

Since a considerable period had elapsed since the first series of tests, the previous activity fungal strains were no longer available, so these oils were tested against two different clinical isolates of *M. mycetomatis*. The sandalwood sample from Table 1 (EO-24) was re-tested against these strains for control, along with the newly obtained oils. The results are reported in Table 2.

All natural EOs tested in this series showed activity against the two fungal strains. Against strain SAK-E07, the highest level of activity was even slightly higher than that observed in the previous tests, i.e., an MIC value of 0.0019% (*v*/*v*) was displayed by two different samples of *S. austrocaledonium* oil (Ess-EO4 and doT-EO9) and by an *S. paniculatum* oil (Ess-EO5), alongside several *S. album* oils from different origins (doT-EO1, doT-EO6 and doT-EO8). Several oils (doT-EO-7, -9 and -11) from *S. album*, *S. paniculatum* and *S. austrocaledonicum* displayed MIC values as low as 0.001% (*v*/*v*). The SW oils doT-EO8 (*S. album*) and doT-EO9 (*S. austrocaledonicum*) were thus the most effective oils against both fungal isolates, with MIC values of 0.0019% and 0.001% (*v*/*v*).

It is to be noted that there appears to be no direct relationship between the *Santalum* species of origin and the oils’ activity against either fungal strain. The standard antifungal drug, itraconazole, was about 20–40 times more active than the most active oils.

In addition to the natural essential oils, the biotechnological product Isobionics^®^ Santalol was also tested; its activity was determined later and is reported in Section 2.5 below. It was comparable to that of the most active natural oils.

### 2.3. Chemical Composition of the Investigated Sandalwood Oils

The chemical composition of all tested SW oils was analyzed by gas chromatography coupled to quadrupole time-of-flight mass spectrometry (GC/QTOF MS). The individual constituents were characterized by their high-resolution mass spectra (HR MS) and linear retention indices (LRIs), which were determined by analyses on two different GC-columns, a DB5 and a WAX column, and identified by library searches (NIST library [10]), as well as a comparison with the literature data [11,12,13,14]. All minor components included, almost 300 different analytes were detected in the 18 EOs analyzed. The full lists of all analytes detected by analysis on the two different columns are reported in Appendix A. The 26 most prominent constituents (based on their percentage in the *S. album* oils) are summarized in Table 3 and Table 4.

These 26 constituents account for ≥90% of the total peak areas in most of the *Santalum* oils analyzed (*S. album*, *S. paniculatum*, *S. austrocaledonicum*) and still constitute 85 and 77% in *S. spicatum* and *S. lanceolatum*, respectively. In contrast, the two “non-*Santalum* sandalwood oils” obtained from the *Amyris* and *Brachylaena* species contain less than 5% of these analytes. As previously known, the sesquiterpenes (*Z*)-α-santalol and (*Z*)-β-santalol represent the major constituents in most cases of *Santalum* oils, except *S. spicatum* and *S. lanceolatum*. While the other *Santalum* species’ EOs contain between 40 and 50% of (*Z*)-α-santalol and 15 to 22% (*Z*)-β-santalol, *S. spicatum* contains much less of these two characteristic sequiterpene alcohols, namely, 21 and 8%, respectively. Significantly higher amounts of (*Z*)-nuciferol and (2*E*,6*E*)-farnesol (11 and 10%, respectively) characterize this species’ oil. *S. lanceolatum* is even poorer in (*Z*)-santalols, with less than 1% of either. It is dominated by (*Z*)-lanceol (31%) and (*Z*)-nuciferol (20%) instead.

Earlier work on GC-MS analyses of SW oils [11,12,13,14,15] had reported a similar pattern of constituents. Overall, to our knowledge, the present report is the most comprehensive comparative analysis of the constituents in the oils of different *Santalum* species.

The two non-*Santalum* EOs (*Amyris balsamifera*, *Brachylaena huillensis*) did not contain any detectable amounts of the santalols. While *A. balsamifera* has a few commonalities in some of its minor constituents with the true SW oils, *B. huillensis* does not contain any of the 26 compounds listed in Table 3 and Table 4. The most abundant constituents in these two oils, according to our analyses (>10% as determined on the WAX column; see Appendix A), were valerianol, γ- and β-eudesmol, constituting 21, 11 and 12% of the *Amyris balsamifera* oil, and ylangenol and α-amorphene, with 12% and 11% in the oil of *Brachylaena huillensis*. Our analyses of these two oils match well with previous findings [16,17]. Regarding *A. balsamifera*, the main constituents identified (>10% [16]) were valerianol, 10-*epi*-γ-eudesmol, guaiol and elemol, which were all recovered in the present analyses, albeit in different quantities. The same applies to components that were detected in smaller quantities (<10% [16]), such as α-curcumene, γ- and β-eudesmol. In addition to the compounds identified previously [16], 14 further components were identified in our analyses, the most abundant being 7-*epi*-α-eudesmol, at around 7%. In previous work on the essential oil of *Brachylaena huillensis*, α-amorphene was found to be the main component, comprising around 17% of the oil. Furthermore, ylangenol and its corresponding aldehydes ylangenal, δ-cadinene and α-calacorene, as well as the two ketoaldehydes 8-ketoylangenal and 8-ketocopaenal, also known as brachylaenalone A and B, were reported to be major components [17]. All the mentioned analytes and 25 more (Appendix A) were found in amounts comparable to those in the previous study. In addition, we found that the sesquiterpene alcohol spirojatamol had a content of 9%, which was not identified in the previous analysis of this oil [17].

The commercial biotechnological product Isobionics^®^ Santalol is a mixture of terpenoids obtained by the fermentation of glucose, utilizing bacteria of the species *Cereibacter sphaeroides* (formerly counted as a *Rhodobacter* species) that has been genetically modified with a gene for santalene synthase [18]. It is specified by the manufacturer to consist of (*Z*)- and (*E*)-α and β-santalols [19]. Our GC-MS analysis showed that in addition to these four isomers, the mixture contains some further constituents in smaller amounts. However, it differs from the natural sandalwood oils mainly by the presence of higher amounts of (*E*)-santalols. Thus, (*E*)-α-santalol commonly constitutes < 1% of the natural oils, but reaches a concentration of almost 16% in the biotechnological product. The (*E*)-β-isomer also is less than 2% in the natural oils but amounts to >7% in the Isobionics^®^ product.

**Table 3 molecules-29-01846-t003:** Main constituents in the investigated sandalwood oils according to GC-QTOF MS analysis on a DB-5 and a DB-HeavyWax (WAX) column. The 26 most abundant components in *S. album* oils and their linear retention indices (LRI), measured on both columns and compared with the literature [11,14,20] and database values, are listed. Each compound was also identified by the match of its QTOF mass spectrum with the corresponding entry in the NIST library [10]; the matching coefficients are included in the table. Their values, expressed in %, represent the best match obtained for each compound in one of the analyses. In a few cases, the compounds were not included in the NIST library but could be identified on grounds of the spectra and LRI values reported in the literature. For a full list of detected constituents in all oils with their LRI values, see Appendix A.

No.	Compound	LRI (WAX)	Lit. [11] (WAX)	NIST (WAX)	LRI (DB-5)	Lit. [14,20] (DB-5)	NIST (DB-5)	Spectra Match (%; DB-5)
1	α-santalene	1583	1575	1576	1425	1416	1420	96.5
2	*epi*-β-santalene	1641	1637	1638	1451	1445	1448	92.9
3	β-santalene	1655	1651	1649	1463	1457	1462	96.9
4	β-bisabolene	1730	1724	1727	1511	1505	1509	96.4
5	β-curcumene	1743	1738	1738	1514	1514	1514	90.0
6	α-curcumene	1777	1770	1777	1485	1479	1483	92.9
7	(*E*)-nerolidol	2044	2023	2042	1564	1561	1564	91.0
8	cyclosantalal	2126	2156	-	1668	-	-	-
9	*epi*-cyclosantalal	2134	2187	-	1686	-	-	-
10	β-bisabolol	2148	2144	2151	1669	1674	1671	92.3
11	(*E*)-α-santalal	2156	2172	2196	1674	-	1679	-
12	α-bisabolol	2212	2207	2215	1685	1685	1684	84.7
13	campherenol	2273	2291	2245	1667	-	1654	-
14	(*Z*)-α-santalol	2339	2344	1873	1673	1674	1681	97.6
15	(*Z*)-*trans*-α-bergamotol	2350	2344	2328	1690	1690	1700	94.8
16	(2*E*,6*E*)-farnesol	2351	2342	2356	1720	1742	1722	78.0
17	(*E*)-α-santalol	2379	2381	-	1695	-	-	-
18	(*Z*)-*epi*-β-santalol	2406	2406	-	1703	1702	1709	96.3
19	(*Z*)-β-santalol	2420	2423	2413	1715	1715	1715	97.8
20	(*Z*)-γ-curcumen-12-ol	2435	2432	-	1711	1728	1711	-
21	(*E*)-*epi*-β-santalol	2448	-	-	1722	-	-	-
22	(*E*)-β-santalol	2460	2465	2465	1736	1738	1741	81.4
23	(*Z*)-β-curcumen-12-ol	2482	2478	-	1752	1754	1746	87.8
24	(*Z*)-lanceol	2487	2484	2518	1757	1760	1763	95.5
25	(*Z*)-nuciferol	2517	2513	2545	1723	1724	1735	92.1
26	spirosantalol	2532	2546	-	1736	-	-	-

**Table 4 molecules-29-01846-t004:** Main constituents in the investigated sandalwood oils and their relative amounts in percent of total peak areas from the GC-QTOF MS total ion chromatograms. The 26 most abundant components in *S. album* oils and their percentage (% of total peak area in the WAX chromatograms) are listed. The color scale indicates the increasing percentage in the form of a heatmap from lowest (0.0, white) to highest (>45, red) values. For a full list of the detected constituents in all oils, see Appendix A.

		*S. album*	*S. album*	*S. album 19*	*S. album (ID)*	*S. album (ID)*	*S. album (IN)*	*S. album (AUS)*	*S. paniculatum*	*S. paniculatum 19*	*S. paniculatum*	*S. austrocaledonicum*	*S. austrocaledonicum*	*S. austrocaledonicum*	*Biotechn. synthetic*	*S. spicatum*	*S. lanceolatum*	*A. balsamifera*	*B. huillensis*
No.	Compound	EO-24	Ess-EO1	doT-EO1	doT-EO5	doT-EO6	doT-EO8	doT-EO11	Ess-EO5	doT-EO2	doT-EO7	doT-EO9	doT-EO10	Ess-EO4	Isobionics	Ess-EO2	Ess-EO3	Ess-EO6	Ess-EO7
1	α-santalene	0.9	1.1	0.5	0.5	0.8	0.5	0.9	1.0	1.1	0.7	1.1	1.0	0.7	3.3	0.6	0.0	0.0	0.0
2	*epi*-β-santalene	1.0	1.1	0.7	0.6	0.8	0.6	1.0	0.9	0.9	0.6	0.9	0.8	0.5	0.2	0.3	0.0	0.0	0.0
3	β-santalene	1.5	1.9	1.1	0.9	1.3	1.0	1.5	1.2	0.9	0.8	0.9	0.8	0.5	1.4	0.5	0.0	0.0	0.0
4	β-bisabolene	0.1	0.0	0.0	0.1	0.1	0.1	0.0	0.2	0.3	0.1	0.2	0.3	0.2	0.1	0.2	1.8	0.7	0.0
5	β-curcumene	0.1	0.1	0.1	0.1	0.0	0.1	0.0	0.2	0.1	0.1	0.2	0.1	0.1	0.0	0.4	0.8	0.0	0.0
6	α-curcumene	0.3	0.3	0.3	0.2	0.3	0.3	0.4	0.3	0.4	0.3	0.3	0.3	0.3	0.0	0.5	0.8	2.0	0.0
7	(*E*)-nerolidol	0.0	0.0	0.0	0.0	0.0	0.0	0.0	0.1	0.1	0.1	0.1	0.1	0.2	0.0	2.7	1.0	0.7	0.0
8	cyclosantalal	0.2	0.2	0.6	0.4	0.3	1.1	0.3	1.4	1.6	1.3	1.6	1.1	0.5	0.0	0.0	0.0	0.0	0.0
9	*epi*-cyclosantalal	0.1	0.1	0.2	0.1	0.1	0.3	0.0	0.3	0.3	0.2	0.3	0.2	0.1	0.0	0.0	0.0	0.0	0.0
10	β-bisabolol	0.4	0.4	0.5	0.5	0.4	0.4	0.4	0.7	0.5	0.7	0.5	0.5	0.5	0.0	2.1	2.0	0.0	0.0
11	(*E*)-α-santalal	1.9	1.4	2.4	1.5	1.4	2.9	2.5	3.0	1.7	3.0	1.4	1.2	0.9	0.2	0.4	0.0	0.0	0.0
12	α-bisabolol	0.1	0.1	0.1	0.2	0.1	0.1	0.1	0.3	0.3	0.2	0.4	0.5	0.6	0.0	7.5	2.7	0.0	0.0
13	campherenol	0.5	0.5	0.7	0.4	0.6	0.5	0.4	0.5	0.3	0.5	0.3	0.3	0.3	0.0	0.4	0.0	0.0	0.0
14	(*Z*)-α-santalol	49.4	48.2	49.3	46.3	48.5	50.7	48.2	42.4	47.0	45.9	47.5	46.1	45.4	31.9	21.1	0.4	0.0	0.0
15	(*Z*)-trans-α-b.^a^	5.8	6.8	4.4	7.1	6.7	4.6	4.1	4.5	5.0	3.4	5.4	5.7	5.9	2.3	3.5	0.0	0.0	0.0
16	(2*E*,6*E*)-farnesol	0.0	0.0	0.0	0.0	0.0	0.0	0.0	t	t	t	t	t	0.4	0.0	10.2	3.3	0.3	0.0
17	(*E*)-α-santalol	0.4	0.4	0.3	0.3	0.3	0.3	0.3	0.2	0.3	0.3	0.3	0.3	0.3	15.7	0.2	0.0	0.0	0.0
18	(*Z*)-*epi*-β-santalol	3.2	3.4	4.5	3.0	3.2	4.0	3.8	3.5	3.5	3.9	3.5	3.3	3.4	5.3	1.7	0.0	0.0	0.0
19	(*Z*)-β-santalol	22.2	22.0	20.2	22.0	20.7	21.1	20.6	16.1	16.7	15.3	18.2	17.3	18.9	18.9	7.9	0.2	0.0	0.0
20	(*Z*)-γ-c.-12-ol ^a^	0.0	0.0	0.0	0.1	0.0	0.0	0.0	1.0	0.3	0.3	0.2	0.2	0.2	0.0	2.8	7.2	0.0	0.0
21	(*E*)-*epi*-β-santalol	0.2	0.2	0.2	0.1	0.2	0.2	0.2	0.1	0.1	0.2	0.1	0.1	0.1	2.1	0.2	0.0	0.0	0.0
22	(*E*)-β-santalol	1.6	1.7	1.1	1.7	1.4	1.2	1.0	1.0	t	1.0	t	t	t	7.3	t	0.0	0.0	0.0
23	(*Z*)-β-c.-12-ol ^a^	0.1	0.2	0.1	0.6	0.2	0.1	0.0	1.7	0.7	0.8	0.7	0.5	0.5	0.0	6.0	6.3	0.0	0.0
24	(*Z*)-lanceol	1.1	1.1	1.6	2.5	2.9	1.7	1.6	3.5	6.8	2.3	6.1	9.2	11.4	0.6	3.0	30.8	0.0	0.0
25	(*Z*)-nuciferol	0.7	1.1	1.5	3.5	2.3	0.9	1.5	5.5	2.7	7.9	2.4	2.1	1.9	0.7	11.3	19.9	0.0	0.0
26	spirosantalol	0.8	1.0	0.8	0.7	1.0	0.7	0.8	2.0	0.5	1.7	0.5	0.6	0.6	0.0	0.8	0.0	0.0	0.0
	total (%)	92.6	93.2	91.4	93.3	93.5	93.4	90.0	91.5	92.8	91.6	93.0	93.0	94.3	90.0	84.5	77.2	3.7	0.0
	(*Z*)-Santalols (%)	71.6	70.2	69.5	68.3	69.2	71.8	68.9	58.5	64.1	61.1	65.6	63.5	64.2	50.8	29.1	0.6	0.0	0.0

^a^ For full names, see Table 3. t: Assignments are tentative and percentages cannot be reported due to peak overlap.

A full list of all detected constituents of the oils analyzed in this study is included in Appendix A.

### 2.4. Separation of Sandalwood Oils and Isolation of (Z)-α-Santalol and (Z)-β-Santalol

To investigate the contribution of the main fractions and constituents of Sandalwood oils to the antifungal activity, the sesquiterpene hydrocarbons and alcohols of Indian and Hawaiian SW oils (EO-24 and Ess-EO5, respectively) were separated by column chromatography (CC) on silica gel 60. In both cases, the column was first eluted with cyclohexane until all constituents moving with this solvent had been eluted, yielding the hydrocarbon fraction (HF1 and HF2, respectively). Subsequently, the column was washed with ethyl acetate to obtain a fraction of sesquiterpene alcohols (AF1 and AF2 from the two oils, respectively). AF1, obtained from EO-24, was further separated on silver-nitrate-impregnated silica gel according to [21] with dichloromethane (DCM) and DCM-methanol mixtures. The two main constituents, (*Z*)-α-santalol and (*Z*)-β-santalol (see Figure 1), were thereby obtained in almost pure form (96.8 and 82.4%, determined by GC-MS).

### 2.5. In Vitro Activity of HF, AF and Isolated (Z)-α-Santalol and (Z)-β-Santalol against M. mycetomatis

The in vitro activity of the separated sesquiterpene hydrocarbon and sesquiterpene alcohol fractions, as well as the purified (*Z*)-santalols, were tested against the *M. mycetomatis* strain SAK-E07 (see Table 5). It is noteworthy that the HF2 (obtained from Ess-EO5) was found to be essentially inactive (MIC > 4512 µg/mL), whereas the AF2 obtained from the same oil showed significant activity, with an MIC value of 64 µg/mL; however, it was somewhat less active than the oil from which it had been obtained (MIC = 16 µg/mL).

The purified santalols differed in activity, (*Z*)-α-santalol being the more active isomer with an MIC value of 125 µmol/L (27.5 µg/mL), while (*Z*)-β-santalol was less active with MIC = 250 µmol/L (55 µg/mL). By comparison at a molar scale, the standard antifungal drug itraconazole (MIC = 0.35 µmol/L) was about 360 times more active than (*Z*)-α-santalol. It should also be noted that even the more active sesquiterpene alcohol was less active than the total oil Ess-EO5. Hence, it cannot be excluded that other constituents, present in lower concentrations, contribute significantly to the activity of the total oil. Until this has been clarified in further studies, the total essential oil remains the best option for further pharmacological investigations.

## 3. Materials and Methods

### 3.1. Tested Materials

All essential oils tested in the initial screening (Table 1, EO-01–EO-27) were of commercial origin. They were obtained from Caesar & Loretz & GmbH, Hilden, Germany (Caelo), except EO-01, which originated from Carl Roth GmbH & Co. KG (Karlsruhe, Germany), EO-26 (purchased in a public Pharmacy in Germany) and EO-27 (Spinnrad GmbH, Gelsenkirchen, Germany). The sandalwood and related oils Ess-EO1–Ess-EO7 (Table 2) were purchased from Essence-pur GmbH (Lohmar, Germany). The sandalwood oils doT-EO1–doT-EO11 were from dōTERRA (Pleasant Grove, UT, USA). The manufacturer generously donated a sample of Isobionics^®^ santalol (BASF SE, Ludwigshafen, Germany). The batch numbers are reported in Table 6.

### 3.2. Chemicals, Sorbents and Solvents

All solvents (*n*-hexane, cyclohexane, dichloromethane (DCM), ethyl acetate (EtOAc), and methanol (MeOH)), as well as silica gel 60 (70–230 mesh) for the column chromatographic (CC) separations, were purchased from Fisher Scientific GmbH (Schwerte, Germany). Solid silver nitrate was purchased from Degussa (Essen, Germany).

### 3.3. Gas Chromatography–Quadrupole Time-of-Flight Mass Spectrometry (GC-QTOF MS)

All analyses were carried out with an Agilent Technologies (Santa Clara, CA, USA) GC-QTOF MS System consisting of a 7890 B gas chromatograph coupled to a 7250 quadrupole time-of-flight (QTOF) mass spectrometer and equipped with a PAL3 RSI Autosampler.

GC parameters: Carrier gas: N_2_; Injector temperature: 220 °C; Injection mode: Split mode; Split ratio: 50:1.

System A: Agilent DB-5 column (diameter 0.25 mm; length 30 m; film thickness 0.25 μm;); Gas flow 1.2 mL/min, linear velocity 41.195 cm/s; Temperature program: initial 120 °C, 2 °C/min to 146 °C, 1 °C/min to 163 °C

System B: Agilent DB-HeavyWax column (diameter 0.25 mm; length 30 m; film thickness 0.25 μm); Gas flow 1.2 mL/min, linear velocity 40.853 cm/s; Temperature program: initial 100 °C, 4 °C/min to 180 °C, 2 °C/min to 220 °C, 4 °C/min to 260 °C

MS parameters: Transfer line temperature: 280 °C; Ion source temperature: 200 °C;

System A: Ionization energy: EI (70 eV); Ion transfer voltages: Lenses 1–5: 9.5 V, −11.9 V, −11.9 V, −11.9 V, 5.3 V respectively; Quadrupole collision cell settings: Temperature 150 °C; mass range 50–1000 m/z; entrance potential 9.0 V; collision energy 0.0 V; exit potential 10.0 V. Time-of-flight (TOF-) voltages: pusher: 1200.0 V; puller: –700.0 V; mirror front: −7000.0 V; mirror mid: –1724.8 V; mirror back: 1175.0 V; detector voltages: microchannel plate: 775 V; photomultiplier 518 V. Data acquisition rate: 3 Spectra/s. System B: Ionization energy: EI (70 eV); Ion transfer voltages: Lenses 1–5: 7.5 V, −13.9 V, –13.9 V, –13.9 V, 5.3 V respectively; Quadrupole collision cell settings: Temperature 150 °C; mass range 50–1000 m/z; entrance potential 9.0 V; collision energy 0.0 V; exit potential 12.0 V. Time-of-flight (TOF-) voltages: pusher: 1200.0 V; puller: –700.0 V; mirror front: –7000.0 V; mirror mid: –1724.8 V; mirror back: 1175.0 V; detector voltages: microchannel plate: 775 V; photomultiplier 531 V. Data acquisition rate: 3 spectra/s.

For the analysis, essential oils were diluted with *n*-hexane at a ratio of 1:2000. The injection volume was 1 µL.

Software used for evaluation: Agilent MassHunter Unknown Analysis (v. 10.0), MassHunter Qualitative Analysis Navigator (v. B.08.00).

The linear retention indices (LRI) were determined automatically by the Software Agilent MassHunter Unknown Analysis.

### 3.4. Separation of Sandalwood Oil to Obtain Sesquiterpene Hydrocarbon and Sesquiterpene Alcohol Fractions

Sandalwood oil, 1.05 g (*Santalum album*, EO-24), was subjected to CC on 200 g of silica gel 60 (column dimensions: inner diameter 60 mm, height 300 mm), eluted with 750 mL of cyclohexane until all of the hydrocarbons eluted. The column was eluted with 750 mL of EtOAc to obtain the sesquiterpene alcohol fraction (AF). After evaporation using the rotary evaporator (25 °C; 200 mbar) followed by further evaporation using the Speedvac (25 °C, 900 r/min), the cyclohexane eluates yielded 48 mg of the sesquiterpene hydrocarbon fraction (HF1) and the ethyl acetate eluates yielded 536 mg of the sesquiterpene alcohol fraction (AF1).

In a second separation, 3.42 g of Hawaiian sandalwood oil (*Santalum paniculatum*, Ess-EO5) was subjected to CC over 200 g of silica gel 60 (70–230 mesh), eluted with 1400 mL of cyclohexane until all of the hydrocarbons eluted. Thereafter, the column was eluted with 1100 mL of EtOAc. After rotary evaporation (25 °C; 200 mbar) followed by further evaporation using the Speedvac (25 °C, 900 r/min), 0.124 g of hydrocarbon fraction (HF2) and 3.39 g of sesquiterpene alcohol fraction (AF2) were obtained.

### 3.5. Isolation of (Z)-α-Santalol and (Z)-β-Santalol

Silica gel 60 (40 g) was suspended in 300 mL of a methanolic AgNO_3_ solution (4% *w*/*v*). After rotary evaporation (45 °C, 330 mbar), 53.47 g of silver-nitrate-impregnated silica gel was obtained. Silica gel 60 (10 g) followed by the silver-nitrate-impregnated silica gel (53.5 g) was packed onto a CC column (column dimensions: inner diameter 20 mm; height 500 mm).

The sesquiterpene alcohol fraction (AF1, 0.536 g), see Section 3.4, was loaded on the column and eluted with 120 mL of DCM, 750 mL of DCM/MeOH (98.5/1.5), and 200 mL of DCM/MeOH (97/3). The eluates were collected in portions of 15 mL. After TLC control on silica 60 F 254 plates, eluates containing similar constituents were combined in 11 fractions. After rotary evaporation (25 °C; 200 bar) and further evaporation using the Speedvac (25 °C, 900 r/min), fraction ZAg2 consisted of 96.80% α-santalol. Furthermore, fraction ZAg5 contained 82.35% β-santalol.

### 3.6. Biological Testing

#### 3.6.1. Cultivation of Various Strains of *M. mycetomatis*

The strains designated SAK-E07 and SAK-E013 used in this study are kept and maintained at the Faculty of Pharmacy Depository, University of Science and Technology (UST), Omdurman, Sudan. The *M. mycetomatis* strains were sub-cultured on Sabouraud Dextrose (SDA) plates and were propagated at 37 °C for 2–3 weeks.

#### 3.6.2. In Vitro Activity Tests with *M. mycetomatis*

The inocula of *M. mycetomatis* strains in RPMI (Roswell Park Memorial Institute) 1640 were sonicated for 10 s (Qsonica, Newtown CT, USA, Q55 sonicator), centrifuged at 2600× *g* for 5 min and incubated for 7 days at 37 °C. After one week, the mycelia were washed and re-suspended in fresh RPMI 1640 medium to give a fungal suspension of 68% to 72% transmission at 660 nm (JENWAY 6305 UV/Vis spectrophotometer).

The minimum inhibitory concentrations (MICs) were determined using the microdilution method [22]. A 1:2 serial dilution of test compounds dissolved in dimethyl sulfoxide (DMSO) was prepared in a 96-well microtiter plate. The *M. mycetomatis* strain(s) were subjected to EOs at concentrations ranging between 0.0019 and 0.25% *v*/*v*, 31.25 and 1000 µM of pure compounds, and 16 and 512 μg/mL of isolated fractions. The standard antifungal agent, itraconazole (Janssen Pharmaceutical Products, Beerse, Belgium), was used as the positive control from 1 to 0.03 μg/mL. To each well, 100 µL of adjusted fungal suspension and 1 µL of the test compound were added, followed by 1 µL of resazurin to give a final concentration of 0.15 mg/mL. The plates were sealed and incubated at 37 °C for 7 days. The assay plates were inspected on day 7 for visual and spectrometric endpoints. For spectrophotometric MICs, absorbance was measured at 600 nm (Thermo Scientific Multiskan Spectrum, Thermo Fischer Scientific, Joensuu, Finland). All assays were performed in three independent replicates [22].

## 4. Conclusions

The sandalwood oil of *Santalum album*, with an MIC value of 0.0039% (*v*/*v*), was found to show the most potent activity among a diverse array of essential oils tested against the most common fungal pathogen of mycetoma, *Madurella mycetomatis*. Various oils of different *Santalum* species and two “Sandalwood” oils obtained from other plants all displayed significant activity against the pathogen. No clear correlation was established between the activity and the species of origin, nor with the component profiles obtained with the GC-MS analyses, in which more than 270 different components were detected in the 18 oils analyzed. However, it could be demonstrated that the antifungal activity is caused by the fraction of sesquiterpene alcohols, while the sesquiterpene hydrocarbons were inactive. Of the two major constituents, (*Z*)-α-santalol and (*Z*)-β-santalol, the former was about twice as potent, but both were less active than the total oil they obtained. It will be interesting to investigate in further studies whether the observed effects are due to an antimitotic effect, as previously postulated [8].

This study thus provides evidence that sandalwood oil, as a natural mixture of volatile sesquiterpenes, has interesting potential as an antimycetomal agent. Further studies on the antifungal potency of the isolated constituents and their possible synergistic effects may reveal structure–activity relationships among these compounds. Furthermore, studies with an in vivo model of mycetoma with infected *Galleria mellonella* larvae have been initiated to assess the possible in vivo efficacy of sandalwood oils and their components. The results of these ongoing experiments will be published separately.

## Figures and Tables

**Figure 1 molecules-29-01846-f001:**
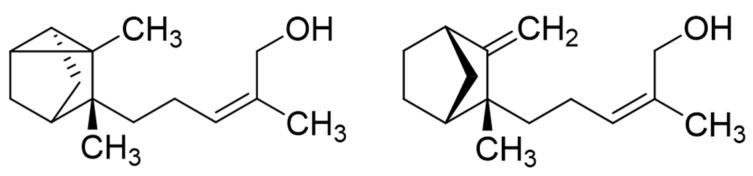
Structures of (*Z*)-α-santalol (left) and (*Z*)-β-santalol (right).

**Table 1 molecules-29-01846-t001:** Antifungal screening of various essential oils against *Madurella mycetomatis*. Data represent average minimum inhibitory concentrations (MIC) in % (*v*/*v*) from four different isolates ± standard deviation (SD).

Code	Essential Oil	Plant of Origin	MIC	±SD
EO-01	Clary Sage	*Salvia sclarea*	0.1068	0.1270
EO-02	Thyme	*Thymus vulgaris/zygis*	0.0244	0.0137
EO-03	(Field) Mint	*Mentha arvensis*	0.0801	0.0565
EO-04	Juniper	*Juniperus communis*	0.0332	0.0224
EO-05	Star Anise	*Illicium verum*	0.1302	0.1173
EO-06	Lemon Balm	*Cymbopogon + Citrus* spp. *	n.a.	
EO-07	Rosemary	*Salvia rosmarinus*	n.a.	
EO-08	Marjoram	*Origanum majorana*	n.a.	
EO-09	Caraway	*Carum carvi*	0.0645	0.0479
EO-10	Patchouli	*Pogostemon cablin*	0.0254	0.0117
EO-11	Cinnamom	*Cinnamomum verum*	0.0127	0.0059
EO-12	Wormwood	*Artemisia absinthium*	0.0488	0.0274
EO-13	Pine Needle	*Pinus sylvestris*	0.0205	0.0133
EO-14	Ginger	*Zingiber officinale*	0.0508	0.0235
EO-15	Clove	*Syzygium aromaticum*	0.0215	0.0117
EO-16	Fir Needle	*Picea abies*	0.1563	0.1326
EO-17	Lavender	*Lavandula* sp.	n.a.	
EO-18	Tea Tree	*Melaleuca alternifolia*	n.a.	
EO-19	Citronella	*Cymbopogon* sp.	0.1302	0.1173
EO-20	Cedar Wood	*Cedrus* sp.	0.0332	0.0224
EO-21	Chamomile	*Matricaria chamomilla*	0.0176	0.0098
EO-22	Basil	*Ocimum basilicum*	0.0801	0.0565
EO-23	Sage (30%)	*Salvia officinalis*	0.0791	0.0581
**EO-24**	**Sandalwood**	***Santalum album***	**0.0039**	**0.0000**
EO-25	Sweet Flag	*Acorus calamus*	0.0215	0.0117
EO-26	Eucalyptus	*Eucalyptus* sp.	0.1133	0.1016
EO-27	Sweet Fennel	*Foeniculum vulgare*	n.a.	
Pos. contr.	itraconazole	*-*	0.0001	0.0000

* Artificial lemon balm or melissa oil is a citronella and lemon oil mixture. n.a.: not active.

**Table 2 molecules-29-01846-t002:** Activity of various sandalwood (SW) oils against two different strains of *Madurella mycetomatis* (clinical isolates). Data represent minimum inhibitory concentrations (MIC) in % (*v*/*v*). In the case of the positive control, the standard antifungal drug itraconazole, the MIC is expressed in % (*m*/*v*). The color scale indicates the increasing activity in the form of a heatmap from the highest (0.0625, white) to lowest (≤0.00005, red) MIC values.

Code	Essential Oil (Species of Origin)	*M. mycetomatis* Strain	
		SAK-E07	SAK-E013
EO-24	Indian SW (*S. album*)	0.0039	0.0039
Ess-EO1	(East) Indian SW (*S. album*)	0.0039	0.0078
Ess-EO2	Australian SW (*S. spicatum*)	0.0039	0.0039
Ess-EO3	North Australian SW (*S. lanceolatum*)	0.0078	0.0078
Ess-EO4	Pacific SW (*S. austrocaledonicum*)	0.0019	0.0078
Ess-EO5	Hawaiian Royal SW (*S. paniculatum*)	0.0019	0.0039
Ess-EO6	West Indian “SW” (*Amyris balsamifera*)	0.0039	0.0078
Ess-EO7	Muhuhu or African “SW” (*Brachylaena huillensis*)	0.0625	0.0078
doT-EO1	doTERRA SW-19 (*S. album*)	0.0019	n.d.
doT-EO2	doTERRA Hawaiian SW-19 (*S. paniculatum*)	0.0039	n.d.
doT-EO5	FPA200727A Indonesian SW (*S. album*)	0.0078	0.0039
doT-EO6	FPA210106A Indonesian SW (*S. album*)	0.0019	0.0039
doT-EO7	KMR210104A Hawaiian SW (*S. paniculatum*)	0.0039	0.001
doT-EO8	KN200902A Indian SW (*S. album*)	0.0019	0.0019
doT-EO9	RO200619A New Caledonian SW (*S. austrocaledonium*)	0.0019	0.001
doT-EO10	RO210112A New Caledonian SW (*S. austrocaledonium*)	0.0039	0.0039
doT-EO11	SAT210122A Australian SW (*S. album*)	0.0078	0.001
Pos. contr.	itraconazole	0.00005	0.000025

n.d. not determined.

**Table 5 molecules-29-01846-t005:** MIC values of Isobionics^®^ Santalol (IsoSant), sandalwood oil Ess-EO5 (*S. paniculatum*) and its separated fractions of sesquiterpene hydrocarbons and alcohols, as well as of the purified (Z)-α- and β-santalols, against the *M. mycetomatis* strain SAK-E07.

Sample	MIC (% *v*/*v*)	MIC (µg/mL)	MIC (µmol/L)
IsoSant		≈27.5 *	≈125 *
Ess-EO5	0.0019	16	
HF2		>512	
AF2		64	
(*Z*)-α-santalol		27.5	125
(*Z*)-β-santalol		55	250
Itraconazole		0.25	0.35

* Approximate values: MIC was determined in µmol/L, assuming the mixture consists exclusively of santalol isomers.

**Table 6 molecules-29-01846-t006:** Origin of all EOs tested in this study.

Code	Essential Oil	Plant of Origin	Manuf.	Batch #
EO-01	Clary Sage	*Salvia sclarea*	Roth	069103645
EO-02	Thyme	*Thymus vulgaris/zygis*	Caelo	72536487
EO-03	(Field) Mint	*Mentha arvensis*	Caelo	77283108
EO-04	Juniper	*Juniperus communis*	Caelo	70816137
EO-05	Star Anise	*Illicium verum*	Caelo	10319008
EO-06	“Lemon Balm” *	*Cymbopogon + Citrus* spp. ***	Caelo	72946467
EO-07	Rosemary	*Salvia rosmarinus*	Caelo	70828477
EO-08	Marjoram	*Origanum majorana*	Caelo	10188603
EO-09	Caraway	*Carum carvi*	Caelo	71717487
EO-10	Patchouli	*Pogostemon cablin*	Caelo	10185505
EO-11	Cinnamon	*Cinnamomum verum*	Caelo	10185204
EO-12	Wormwood	*Artemisia absinthium*	Caelo	72317367
EO-13	Pine Needle	*Pinus sylvestris*	Caelo	70295067
EO-14	Ginger	*Zingiber officinale*	Caelo	63055487
EO-15	Clove	*Syzygium aromaticum*	Caelo	10127104
EO-16	Fir Needle	*Picea abies*	Caelo	72469417
EO-17	Lavender	*Lavandula* sp.	Caelo	73411058
EO-18	Tea Tree	*Melaleuca alternifolia*	Caelo	71984507
EO-19	Citronella	*Cymbopogon* sp.	Caelo	71497257
EO-20	Cedar Wood	*Cedrus* sp.	Caelo	10118703
EO-21	Chamomile	*Matricaria chamomilla*	Caelo	10318402
EO-22	Basil	*Ocimum basilicum*	Caelo	10229308
EO-23	Sage (30%)	*Salvia officinalis*	Caelo	71650068
EO-24	Sandalwood (SW)	*Santalum album*	Caelo	10057806
EO-25	Sweet Flag	*Acorus calamus*	Caelo	96115283
EO-26	Eucalyptus	*Eucalyptus* sp.	Pub. **	**
EO-27	Sweet Fennel	*Foeniculum vulgare* var. *dulce*	Spinnrad	5552
Ess-EO1	Indian SW	*Santalum (S.) album*	Essence	0020040
Ess-EO2	Australian SW	*S. spicatum*	Essence	7300
Ess-EO3	Australian (North) SW	*S. lanceolatum*	Essence	HG CONS 42
Ess-EO4	Pacific SW	*S. austrocaledonicum*	Essence	836094
Ess-EO5	Hawaiian Royal SW	*S. paniculatum*	Essence	HA.07272015
Ess-EO6	West Indian “SW”	*Amyris balsamifera*	Essence	40017
Ess-EO7	African “SW” (Muhuhu)	*Brachylaena huillensis*	Essence	131BB501
doT-EO1	SW-19	*S. album*	dōTERRA	171384A
doT-EO2	Hawaiian SW-19	*S. paniculatum*	dōTERRA	162885
doT-EO5	Indonesian SW	*S. album*	dōTERRA	FPA200727A
doT-EO6	Indonesian SW	*S. album*	dōTERRA	FPA210106A
doT-EO7	Hawaiian SW	*S. paniculatum*	dōTERRA	KMR210104A
doT-EO8	Indian SW	*S. album*	dōTERRA	KN200902A
doT-EO9	New Caledonian SW	*S. austrocaledonicum*	dōTERRA	RO200619A
doT-EO10	New Caledonian SW	*S. austrocaledonicum*	dōTERRA	RO210112A
doT-EO11	Australian SW	*S. album*	dōTERRA	SAT210122A
IsoSant	Isobionics Santalol	Biotechnological product	BASF	2207IB02SALiso

* This is an artificial lemon balm or melissa oil consisting of a mixture of citronella and lemon oil. ** This oil was purchased from a public pharmacy in Germany; it was of Ph. Eur. Quality. The batch number is not known.

## Data Availability

The raw data of all analyses supporting the conclusions of this article are available from the corresponding author (T.J.S.) upon request.

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
