# Peer review of "Sandalwood Oils of Different Origins Are Active In Vitro against Madurella mycetomatis, the Major Fungal Pathogen Responsible for Eumycetoma"

_molecules, 2024, doi:10.3390/molecules29081846_

Round 1

Reviewer 1 Report

Comments and Suggestions for Authors

This paper focused on the Sandalwood oils of different origins are active in vitro against Madurella mycetomatis, the major fungal pathogen responsible for eumycetoma. It is recommended to revise it carefully and review it again.

1. Santalum album EO has been confirmed to have antibacterial activity against M. mycetomatis. Why should we conduct activity screening?

2. What are the practical implications of choosing commercially available essential oils for research?

3. What is the basis for choosing these 27 essential oils?

4. In page 3, Line no.100. Replace ‘sd’ with ‘SD’.

5. Specific MIC concentration values are missing from Table 1.

6. Is it representative that the previous strain is no longer available and a new strain is selected for antibacterial testing? It is recommended to still use the strains used previously, so that the experimental results are more representative.

7. In page 8, Line no.261. Replace ‘µmol/L’ with ‘µg/mL’.

Comments on the Quality of English Language

Minor editing of English language required

Author Response

Reviewer 1

This paper focused on the Sandalwood oils of different origins are active in vitro against Madurella mycetomatis, the major fungal pathogen responsible for eumycetoma. It is recommended to revise it carefully and review it again.

  1. Santalum albumEO has been confirmed to have antibacterial activity against M. mycetomatis. Why should we conduct activity screening?

Reply: Madurella mycetomatis is a fungus, not a bacterium. The essential oils tested, including Sandalwood oil, were not previously tested against this fungus.

  1. What are the practical implications of choosing commercially available essential oils for research?

Reply: The practical implications are that these oils are available and accessible in unvarying and controlled quality. The chosen approach is, therefore, entirely sustainable.

  1. What is the basis for choosing these 27 essential oils?

Reply: The basis for choosing these 27 oils is that they vary widely in the composition of their known constituents, thus warranting a wide array of chemicals tested against the fungus under study. This was a helpful approach since a relatively strongly active oil (sandalwood oil) was found.

  1. In page 3, Line no.100.Replace ‘sd’ with ‘SD’.

Reply: Although we are unaware of an officially prescribed abbreviation for standard deviation, we have changed it to follow the reviewer.

  1. Specific MIC concentration values are missing from Table 1.

Reply: It is not clear what the reviewer means. The MIC is expressed as mean +/- SD of four independently determined values. This should sufficiently demonstrate their activity level against the fungus under study.

  1. Is it representative that the previous strain is no longer available and a new strain is selected for antibacterial testing? It is recommended to still use the strains used previously, so that the experimental results are more representative.

Reply: We would like to emphasize that we investigated antifungal, not antibacterial activity. Indeed, it may happen that clinical isolates (i.e., cultivated fungi obtained from live human patients’ infected tissue) are not immortal (like many “laboratory strains” are). It is, therefore, not possible to use previous strains because they do not exist anymore.

  1. In page 8, Line no.261. Replace ‘µmol/L’ with ‘µg/mL’.

Reply: No, using the molar scale here is correct since we are comparing the potency of single compounds (Santalol and Itraconazole), which should always be done on a molar scale. However, we detected some typographic errors in the last line of the table and in line 261, which were corrected.

Reviewer 2 Report

Comments and Suggestions for Authors

Table 1. Antifungal screening does not contain a standard antifungal agent for comparison. please add.

formula of (Z)-α-Santalol and (Z)-β-Santalol  would be useful. Also spectroscopic data for the verification can be included within the supplementary material for this valuable oil.

The antimic acitivity should also be justified that it should be avoided to use by layman for therapeutical purposes.

The literature should be primary resources and more discussion needs to be included prior final acceptance.

Did the authors get approval/consent from the commercial companies to publish their analysis details with their brands?

The conclusion part of the mans is a repetition of the resulsts in abstract form. Thus, a sound scientific CONCLUSION instead of an extended abstract...  

line 399 "Furthermore, studies with an in vivo model of mycetoma with infected Galleria mellonella larvae have been initiated to assess the possible in vivo efficacy of sandalwood oils and their components" 

where is the exp data for this statement? this is not a conclusion it is a new finding? if not from literature? 

Author Response

Reviewer 2

Table 1. Antifungal screening does not contain a standard antifungal agent for comparison. please add.

Reply: The standard antifungal itraconazole was used as in table 2. It was added to table 1.

formula of (Z)-α-Santalol and (Z)-β-Santalol  would be useful.

Reply: The structures of both santalols were added as Figure 1.

Also spectroscopic data for the verification can be included within the supplementary material for this valuable oil.

Reply: The mass spectra of both isolates have been added to the Supplementary Materials.

The antimic acitivity should also be justified that it should be avoided to use by layman for therapeutical purposes.

Reply: It is not fully clear what the reviewer means. We do not make any claims for therapeutic use but denote that in vivo studies are necessary and have been initiated.

The literature should be primary resources and more discussion needs to be included prior final acceptance.

Reply: We cited primary resources wherever they existed. We considered the discussion as detailed as necessary but tried to be as concise as possible to avoid unnecessary speculations. If this is not sufficient, we ask for clear and detailed instructions on what to cite and/or discuss.

Did the authors get approval/consent from the commercial companies to publish their analysis details with their brands?

There is no legal requirement for such consent.

The conclusion part of the mans is a repetition of the resulsts in abstract form. Thus, a sound scientific CONCLUSION instead of an extended abstract...  

Reply: The first paragraph concisely summarizes the results that we find necessary to understand the following conclusion. The second paragraph provides an obvious conclusion (namely, that “sandalwood oil, as a natural mixture of volatile sesquiterpenes, has an interesting potential as antimycetomal agent”), followed by suggestions for further research (which is also a conclusion from the present results).

line 399 "Furthermore, studies with an in vivo model of mycetoma with infected Galleria mellonella larvae have been initiated to assess the possible in vivo efficacy of sandalwood oils and their components" where is the exp data for this statement? this is not a conclusion it is a new finding? if not from literature? 

Reply: These experiments are ongoing, as indicated by the word “initiated,” meaning the data will be published once the study is finished. We tried to clarify this by adding, “The results of these ongoing experiments will be published separately.”

Reviewer 3 Report

Comments and Suggestions for Authors

Title could be improved as:

Sandalwood oils of different origins are active in vitro against Madurella mycetomatis, one of the fungi responsible for eumycetoma in humans 

L20: delete 'neglected'

L24:  delete 'among the tested EOs'

L71: 'Treatment options for M. mycetomatis infections seem non-existent, with even several azoles, such as  itraconazole, ....showing no promising clinical outcomes' 

Reformulate  as 'Treatments....are currently used.with several azoles,....but they have shown no...'

L84: 'desperately' delete

L86: delete 'in some instances'

L93: delete 'against Madurella mycetomatis'

L112: delete 'against Madurella mycetomatis'

L222: 'The sandalwood sample from Table 1 (EO-24) together with the newly introduced oils were tested against these strains ' 

L134: To reformulate as 'No direct correlation was found between the origin of Santalum species  and the activity of the oils against any of the fungal strains'.

L147: Reformulate: Chemical Composition of the Sandalwood Oils

L148 to L153 '...with literature data [11, 12, 13, 14]' . Better in materials and methods

Reformulate L196 to L197: 'The biotechnological product Isobionics® Santalol is a mixture of ...'

2.4. Separation of Sandalwood Oils and Isolation of (Z)-α-Santalol and (Z)-β-Santalol

It is better to move this paragraph to materials and methods

L258: delete 'somewhat'

L263: Reformulate 'Hence, other constituents in lower concentrations might  contribute to the efficacy of the total oil, which remains the best for  further pharmacological investigations.'

L386: 'obtained' with 'extracted'

L387- L390: Reformulate 'No clear correlation was established between the activity and the species of origin, nor with the component profiles performed with the GC-MS analyses, in which more than 270 different components were detected in the 18 oils analysed.'

L390: to reformulate as: However, it appears that the antifungal activity was due to the sesquiterpene alcohol fraction rather than the sesquiterpene hydrocarbons, which were inactive. The (Z)-α-santalol component was about twice as potent as the (Z)-β-santalol, but both were less active than the whole oil.

L394: To move to the discussion 'It will be interesting to investigate in further studies whether the observed 394 effects are due to an antimitotic effect as previously postulated [8].

L394: Move to the discussion 'Further studies on the antifungal potency of isolated constituents and their possible synergistic effects may reveal structure-activity relationships among these compounds.'

In the materials and methods: the methodology for determining the MIC (Minimum Inhibitory Concentration) was not specified. In general, there is a formula based on the control (growth of the fungus without EO) and its growth in a medium containing increasing concentrations of EO. If there is another technique for determining MIC, it should be detailed.

Author Response

Reviewer 3

Title could be improved as:

Sandalwood oils of different origins are active in vitro against Madurella mycetomatis, one of the fungi responsible for eumycetoma in humans 

Reply: It is indisputably not just “one of the”, but the major fungal pathogen responsible for the disease. To our knowledge, Eumycetoma is a human disease, so mentioning it does not appear necessary. Hence, we do not find it necessary to change the title.

L20: delete 'neglected'

Reply: Mycetoma is one of the neglected tropical diseases defined by the WHO. We do not see why this should go unmentioned.

L24:  delete 'among the tested EOs'

Reply: This was deleted.

L71: 'Treatment options for M. mycetomatis infections seem non-existent, with even several azoles, such as  itraconazole, ....showing no promising clinical outcomes' 

Reformulate  as 'Treatments....are currently used.with several azoles,....but they have shown no...'

Reply: We have changed the wording to: “Current treatment options for M. mycetomatis infections including several azoles, such as itraconazole, fluconazole, posaconazole, voriconazole, ravuconazole, as well as other classes of antifungals, including echinocandins (e.g. caspofungin, anidulafungin, micafungin), allylamines (e.g. terbinafine), and polyenes (e.g. amphotericin B) show no promising clinical outcomes.”

L84: 'desperately' delete

Reply: It was replaced by “urgently” to sound less pathetic.

L86: delete 'in some instances'

Reply: Only some EOs (not all) have previously been tested in a few publications (=some instances). We do not see why this should be deleted.

L93: delete 'against Madurella mycetomatis'

Reply: No, because it is not general antifungal activity but specifically against this particular fungus. We instead deleted “antifungal”.

L112: delete 'against Madurella mycetomatis'

Reply: No, because it is not a general activity but specifically against this particular fungus, which should be mentioned here.

L222: (The reviewer means L122) 'The sandalwood sample from Table 1 (EO-24) together with the newly introduced oils were tested against these strains ' 

Reply: This was reformulated: “The sandalwood sample from Table 1 (EO-24) was re-tested against these strains for control along with the newly obtained oils.” This better reflects the original meaning, which was somehow lost in the submitted version.

L134: To reformulate as 'No direct correlation was found between the origin of Santalum species and the activity of the oils against any of the fungal strains'.

Reply: The wording is focused on the various oils’ species of origin (i.e. different Santalum species) and not so much on these species’ (geographic) origin. The sentence should remain unchanged.

L147: Reformulate: Chemical Composition of the Sandalwood Oils

Reply: We describe only the investigated Sandalwood Oils' chemical composition. Therefore, we do not see why this should be reformulated.

L148 to L153 '...with literature data [11, 12, 13, 14]' . Better in materials and methods

Reply: The relevant information is in the materials and methods. However, it should also be allowed to cite the references in the main text.

Reformulate L196 to L197: 'The biotechnological product Isobionics® Santalol is a mixture of ...'

Reply: We agree that this can be shortened. It now reads, “The commercial biotechnological product Isobionics® Santalol is a mixture of terpenoids obtained by fermentation of glucose utilizing bacteria…”.

2.4. Separation of Sandalwood Oils and Isolation of (Z)-α-Santalol and (Z)-β-Santalol

It is better to move this paragraph to materials and methods

Reply: The reviewer does not specify why it should be better to move this paragraph. The isolation details are in the M&M section, but we disagree with moving the paragraph describing the results.

L258: delete 'somewhat'

Reply: OK, deleted.

L263: Reformulate 'Hence, other constituents in lower concentrations might  contribute to the efficacy of the total oil, which remains the best for  further pharmacological investigations.'

Reply: We do not understand why our original formulation should be paraphrased in this way. We prefer our original wording.

L386: 'obtained' with 'extracted'

Reply: The oils are obtained from the plant materials, usually by hydrodistillation, not “extracted” (which would involve an extraction solvent).

L387- L390: Reformulate 'No clear correlation was established between the activity and the species of origin, nor with the component profiles performed with the GC-MS analyses, in which more than 270 different components were detected in the 18 oils analysed.'

Reply: We adopted the reviewer’s suggested wording.

L390: to reformulate as: However, it appears that the antifungal activity was due to the sesquiterpene alcohol fraction rather than the sesquiterpene hydrocarbons, which were inactive. The (Z)-α-santalol component was about twice as potent as the (Z)-β-santalol, but both were less active than the whole oil.

Reply: From the presented data, it does not only “appear” but is a clear finding. We do not find it helpful to change this wording. However, we have combined the last two statements into one sentence, which now reads: “Of the two major constituents, (Z)-α-santalol and (Z)-β-santalol, the former was about twice as potent, but both were less active than the total oil they obtained.”

L394: To move to the discussion 'It will be interesting to investigate in further studies whether the observed 394 effects are due to an antimitotic effect as previously postulated [8].

L394: Move to the discussion 'Further studies on the antifungal potency of isolated constituents and their possible synergistic effects may reveal structure-activity relationships among these compounds.'

Reply: The Conclusions may also indicate important aspects of future research. We do not find it useful to truncate our conclusions.

In the materials and methods: the methodology for determining the MIC (Minimum Inhibitory Concentration) was not specified. In general, there is a formula based on the control (growth of the fungus without EO) and its growth in a medium containing increasing concentrations of EO. If there is another technique for determining MIC, it should be detailed.

Reply: We have added the following sentence (line 374) to clarify where the method was described. “The minimum inhibitory concentrations (MICs) were determined using the microdilution method [22]”. We believe this is sufficient.

Reviewer 4 Report

Comments and Suggestions for Authors

The authors of the manuscript 'Sandalwood oils of different origins are active in vitro against Madurella mycetomatis, the major fungal pathogen responsible for eumycetoma' present an interesting work on the use of different essential oils in in vitro tests against Madurella mycetomatis. The authors characterised the essential oils and separated sesquiterpene hydrocarbon fractions and alcohols. (Z)-α- and (Z)-β-santalol were isolated and used for in vitro activity tests; (Z)-α-santalol presented the best MIC. Unfortunately, the standard antifungal drug is still the most active. The manuscript is clearly written, and the results are interesting, but contains a small part that needs to be improved before it can be accepted.

Points that need to be addressed:

lines 82, 93, 95, 98, 112 and 141: Please, the fungus should be written M. mycetomatis because it has been already cited in introduction.

Table 1: Why did the authors not include the positive control (itraconazole)? It should only be included to get an idea of the drug susceptibility of the first clinical isolate and then to have data to compare with Table 2. Literature data are also ok.

Line 106: Santalum album can be written as S. album. It has already been cited in the text.

Lines 117-118: Please use short binomial names.

Table 2: Please use short binomial names for all plants.

Lines 179-180: Please use short binomial names.

Table 4: It is probably appropriate to specify the color scale in the legend.

Lines 383-385: Please, use the short binomial name.

Line 387: The sentence 'There did not seem to be a clear correlation of the activity with the species of origin' should be rewritten better, it is not easily understandable. The concept is much better explained in the Results and Discussion section. If necessary, even two or three lines can be used, the important thing is to explain the concept well.

Author Response

Reviewer 4

The authors of the manuscript 'Sandalwood oils of different origins are active in vitro against Madurella mycetomatis, the major fungal pathogen responsible for eumycetoma' present an interesting work on the use of different essential oils in in vitro tests against Madurella mycetomatis. The authors characterised the essential oils and separated sesquiterpene hydrocarbon fractions and alcohols. (Z)-α- and (Z)-β-santalol were isolated and used for in vitro activity tests; (Z)-α-santalol presented the best MIC. Unfortunately, the standard antifungal drug is still the most active. The manuscript is clearly written, and the results are interesting, but contains a small part that needs to be improved before it can be accepted.

Points that need to be addressed:

lines 82, 93, 95, 98, 112 and 141: Please, the fungus should be written M. mycetomatis because it has been already cited in introduction.

Reply: We agree on lines 82 and 95. In other instances, the full name should be written because the titles and headers should be able to stand alone without abbreviations.

Table 1: Why did the authors not include the positive control (itraconazole)? It should only be included to get an idea of the drug susceptibility of the first clinical isolate and then to have data to compare with Table 2. Literature data are also ok.

Reply: The missing data was added to the table.

Line 106: Santalum album can be written as S. album. It has already been cited in the text.

Reply: OK, this has been changed.

Lines 117-118: Please use short binomial names.

Reply: The correct names have been used.

Table 2: Please use short binomial names for all plants.

Reply: The correct names have been used.

Lines 179-180: Please use short binomial names.

Reply: The correct names have been used.

Table 4: It is probably appropriate to specify the color scale in the legend.

Reply: We added a description to the figure caption: “The color scale indicates increasing percentage in the form of a heatmap from lowest (0.0, white) to highest (>45, red) values.”

Lines 383-385: Please, use the short binomial name.

Reply: The correct names have been used.

Line 387: The sentence 'There did not seem to be a clear correlation of the activity with the species of origin' should be rewritten better, it is not easily understandable. The concept is much better explained in the Results and Discussion section. If necessary, even two or three lines can be used, the important thing is to explain the concept well.

Reply: We have already adopted another reviewer’s suggested wording and hope this reviewer can accept it.

Reviewer 5 Report

Comments and Suggestions for Authors

 Dear  Authors, 

there are many results presented and this is very hard to combine in one  MS - maybe more focused would be in making two instead of one  MS. 

 My suggestions are given in the comments...

 Very interesting and easy to read work !!!

 Well done

 Reviewer

Author Response

Reviewer 5

Dear  Authors, 

there are many results presented and this is very hard to combine in one  MS - maybe more focused would be in making two instead of one  MS. 

My suggestions are given in the comments...

Very interesting and easy to read work !!!

Well done

Reply: We thank the reviewer for the positive assessment and for the various suggestions in the annotated pdf. We have taken this into account wherever we found it appropriate. See the responses in the pdf.

Round 2

Reviewer 1 Report

Comments and Suggestions for Authors

You have revised the manscript according to the reviewer's comments. 

Comments on the Quality of English Language

The language need minor edit.